# Health knowledge, health behaviors and attitudes during pandemic emergencies: A systematic review

**Fabio Alexis Rincón Uribe**[1☉]*, **Rejane Célia de Souza Godinho**[2☉], **Marcos Antonio Santos Machado**[2☉], **Kleber Roberto da Silva Gonçalves Oliveira**[2], **Cristian Ariel Neira Espejo**[2,3], **Natalia Carvalho Vianna de Sousa**[2‡], **Leonardo Lucas de Sousa**[2‡], **Marcos Vinicius Monteiro Barbalho**[4‡], **Pedro Paulo Freire Piani**[5], **Janari da Silva Pedroso**[2,6]

**1** Programa de Pós-graduação em Psicologia, Universidade Federal do Pará, Belém, Brasil, **2** Programa de Pós-graduação em Teoria e Pesquisa do Comportamento, Universidade Federal do Pará, Belém, Brasil, **3** Escuela de Psicología, Facultad de Ciencia Sociales y Comunicación, Universidad Santo Tomás, Santiago, Chile, **4** Programa de Graduação em Psicologia, Universidade Federal do Pará, Belém, Brasil, **5** Instituto de Ciências da Saúde, Universidade Federal do Pará, Belém, Brasil, **6** Bolsista Produtividade CNPq - Nível 2, Programa de Pós-graduação em Psicologia, Universidade Federal do Pará, Belém, Pará, Brazil

☉ These authors contributed equally to this work.
‡ These authors also contributed equally to this work.
* faru1095@gmail.com

**Data Availability Statement:** All relevant data are within the manuscript and its Supporting information files.

## Abstract

### Background

People with good health knowledge present a conceptual and objective appropriation of general and specific health topics, increasing their probability to express health protection and prevention measures. The main objective of this study was to conduct a rapid systematic review about the effects of health knowledge on the adoption of health behaviors and attitudes in populations under pandemic emergencies.

### Methods

A systematic review was performed according to PRISMA checklist and the Cochrane method for rapid systematic reviews. Studies searches were performed in APA PsycNet, Embase, Cochrane Library and PubMed Central. Studies published between January 2009 and June 2020 and whose primary results reported a measure of interaction between health knowledge, health attitudes and behaviors in population groups during pandemics were included. A review protocol was recorded in PROSPERO (CRD42020183347).

### Results

Out of a total of 5791 studies identified in the databases, 13 met the inclusion criteria. The included studies contain a population of 26099 adults, grouped into cohorts of health workers, university students, clinical patients, and the general population. Health knowledge has an important influence on the adoption of health behaviors and attitudes in pandemic contexts.

**Funding:** This study was supported by Universidade Federal do Pará, Brasil / Pró-Reitoria de Pesquisa e Pós-graduação (PROPESP) in the form of material and virtual resources (bibliographic material and an institutional license for unlimited access to databases) and funding for the article publication charges provided to FARU, RCdSG, MASM, CANE, LLdS, NCVdS, MVMB, PPFP and JdSP, by Mineração Paragominas S/A (Norsk Hydro Brasil) in the form of scientific research funding awarded to FARU, and by Hospital Universitário Bettina Ferro de Souza in the form of human resources (review and academic guidance by health professionals) provided to KRdSGO and JdSP. The funders had no role in study design, data collection and analysis, decision to publish, or preparation of the manuscript.

**Competing interests:** The authors have declared that no competing interests exist.

## Conclusions

The consolidation of these preventive measures favors the consolidation of public rapid responses to infection outbreaks. Findings of this review indicate that health knowledge notably favors adoption of health behaviors and practices. Therefore, health knowledge based on clear and objective information would help them understand and adopt rapid responses to face a pandemic.

## Introduction

Health knowledge is a theoretical construct that includes detailed and specific information about etiology, prevalence, risk factors, prevention, transmission, symptomatology and disease treatment, as well as on health services and patient rights [1]. These categories characterize an objective nature, since this information is acquired through authorized external sources and therefore can be considered explicit and factual [1, 2]. Previous evidence has demonstrated the positive effects of appropriate levels' health knowledge by community in general in health promotion and disease prevention [3, 4]. Likewise, during public health emergencies, health knowledge reported by the public plays an important role in reducing risky behaviors and adopting protective and preventive practices [5]. However, in the broad empirical research in public health, a consistent and clear definition of health knowledge has not been offered [1] and its lack of conceptual clarity has not allowed solid evidence that show its effects on behavior, or people's attitudes, or their contribution to the development of public policies [6].

In the pandemics' context, the literature evidence suggests that people who report high levels of knowledge about pandemics (definition, development, severity) linked to some aspects of the disease (transmission, prevention, causes) present an increase in the preventive measures practice, such as hand washing, the use of biosafety elements (masks, gloves, disinfectants), avoiding crowds, covering the face when coughing or sneezing, and seeking medical help in case of symptoms [7, 8]. Likewise, adequate knowledge about pandemics and general health information helps to correct misconceptions about circumstances related to pandemics, increasing the perception of susceptibility against the infection risk and improving the self-efficacy of self-protection [9, 10]. However, there are some inconsistencies in the field of health knowledge research during pandemic periods. The knowledge evaluation seems to focus on explicit and general information people have about knowledge's definition, causes and prevention of a pandemic, without investigating if this knowledge is linked, at the same time, with individual and public appropriation on the general and specific issues in health, associated with a personal commitment to take care of themselves and contribute to collective care [11, 12]. Thus, the investigations have focused on the exclusively cross-sectional analysis of these results and have not explored if knowing about general, specific, and objective health information could influence in the adoption of behaviors, practices or attitudes prevention and protection during the infection period and after that [13, 14].

Previous evidence synthesis has addressed the knowledge role in behavioral responses during pandemics. A systematic review of studies published up to 2010 examined the demographic and attitudinal determinants of protective behaviors during a pandemic [15]. In their results, an association was identified between knowledge about the transmission of SARS, and the pandemic's meaning, and a greater execution of prevention behaviors, and intentions to comply with quarantine. Another studies review published between March 2009 and August 2011 analyzed the community response to the H1N1 influenza pandemic, particularly

determining if these behaviors were related to the level of pandemic knowledge [16]. In their results, it is indicated that pandemic knowledge (transmission and prevention) is a factor that contributes to the adoption of preventive behaviors (hygiene behaviors, quarantine compliance, avoiding crowds and wearing a mask). These evidence syntheses have demonstrated the relevant role of knowledge about some health components such as transmission, prevention and the meaning of a pandemic in people's prevention practices. However, is still needed to identify if there are other categories included in the broad conceptual structure of health knowledge, which may be related to adoption of behaviors and attitudes by the public during these health emergencies.

Promotion of health knowledge is a fundamental strategy to maintain people's health during public health emergencies, so, an adequate health knowledge could help communities to understand risk factors and generate rapid responses to contain infection outbreaks [17]. In this way, in the midst of the public health emergency generated by COVID-19, political agents and health entities require synthesis of scientific evidence for the development of plans and strategies that contribute to the consolidation of preventive measures for the mitigation and control of massive outbreaks in all populations, especially those with a more precarious health system. Thus, under this context, the WHO [18] recommends execution of rapid reviews whose results can strengthen health policies and systems. These findings may offer an evidence-based explanatory framework that facilitates the early configuration of rapid behavioral and cognitive responses that guarantee the personal and collective care of communities in the context of pandemic emergencies. In this way, in response to the strategic objectives of the emergency committee, which seek to promote preventive measures and raise awareness of health issues in all populations in the current public emergency of COVID-19 [19], our objective was to offer a rapid synthesis of the evidence about effect of health knowledge, as broader measure that includes detailed and objective information on health components, to adopt health behaviors and attitudes during pandemics.

## Methods

This systematic review was carried out under the standards established in "Preferred Reporting Items for Systematic Reviews and Meta-Analyzes" (PRISMA) [20]. Likewise, the methodological guide "The Interim Guidance from the Cochrane Rapid Reviews Methods Group" was used to conduct the systematic review process [21]. A review protocol was recorded in the International Prospective Registry of Systematic Reviews—PROSPERO (Record number: CRD42020183347). The PRISMA checklist was used to inform this work (S1 Appendix).

### Search strategy

Systematic literature searches were performed on APA PsycNet (American Psychological Association), Embase, Cochrane Library, and PubMed Central. The search criteria included controlled vocabulary MeSH (Medical Subject Headings) on *Health Awareness*, *Health Knowledge*, *Pandemic*, *Health Behavior* and *Health Attitudes*. The search strategy was developed and adapted for all databases. The complete search strategy is available in S2 Appendix.

### Study selection

In this systematic review, studies examining the influence of health knowledge about adoption of healthy behaviors and attitudes during pandemics were included. Likewise, were included studies that conceptually addressed or operationalized interest results as follow: a) Health knowledge, defined as a variable that comprises detailed, explicit and objective information on health categories such as etiology, prevalence, risk factors, prevention, transmission, symptoms

and disease treatment, as well as health services and patient rights [1]; b) Health behavior, understood as any conduct performed by a person to protect or promote health, as well as prevent the appearance of a disease or detect it in an asymptomatic stage [22]; c) Health Attitudes, described as a predisposition to adopt and maintain self-care or prevention practices, favoring adequate perceptions of health [23]. In addition, articles published between January 2009 and June 2020 were included and limited to the English language only. The details of the inclusion and exclusion criteria are available in S1 Table. All references were imported into the Mendeley software to manage data and eliminate duplicates. Three authors independently screened abstracts and full texts of potential studies using the inclusion and exclusion criteria and any discrepancies were discussed in group.

## Data extraction

Data were extracted using a standardized extraction form, and the compilers indicated information about author(s), year, country, design, participant characteristics and findings (primary results and measure of association). For the measures of association, linear dependence indicators (correlation coefficients), odds ratio (Odds Ratio), standardized coefficients and confidence intervals were considered. Likewise, statistical estimates tests were included as solid evidence of the reported findings. The data was extracted by one author with complete verification of two authors.

## Quality assessment

The quality assessment was conducting by one author (FR), with complete verification of JP and KS, using a modified version of the Ottawa–Newcastle scale (ONS) for cross sectional studies [24]. The ONS contains a checklist consisting of three criteria: selection (representativeness of individuals); comparability (determination of confusion) and result (evaluation and analysis of results). Since the ONS does not provide clearly defined and standardized cutoff points, the scores were established considering previous published studies [25, 26]. According to the scoring system, studies are rated in a range of 0 to 10 points and are classified as low (10 and 9 points), medium (7 and 8 points) high (<7 points). In general, the ONS scale has presented good reliability between evaluators and test-retest [27].

## Synthesis and analysis

In this rapid review, findings were presented and synthesized using a narrative and thematic synthesis approach. In this way, thematic categories were created to analyze the effect of health knowledge on adoption of healthy behaviors and attitudes during pandemics. To consolidate these results, statistical estimates representing the magnitude of association between variables that operationalize health knowledge were considered: (transmission, symptoms, prevention, etiology, pandemic definition, vaccine availability, infection severity, incubation period and communicability), behaviors (prevention practices, self-protection, and hygiene behaviors) and health attitudes (perceptions about infection severity and beliefs about pandemics).

## Results

Searching the online databases identified 5791 publications. Also, 5 additional studies were identified, using manual searches through checking the reference lists of included studies and a complementary research on Google Academic. After removing duplicates and selecting abstracts and titles, 92 articles were chosen for screening. Of this total, 79 did not meet the

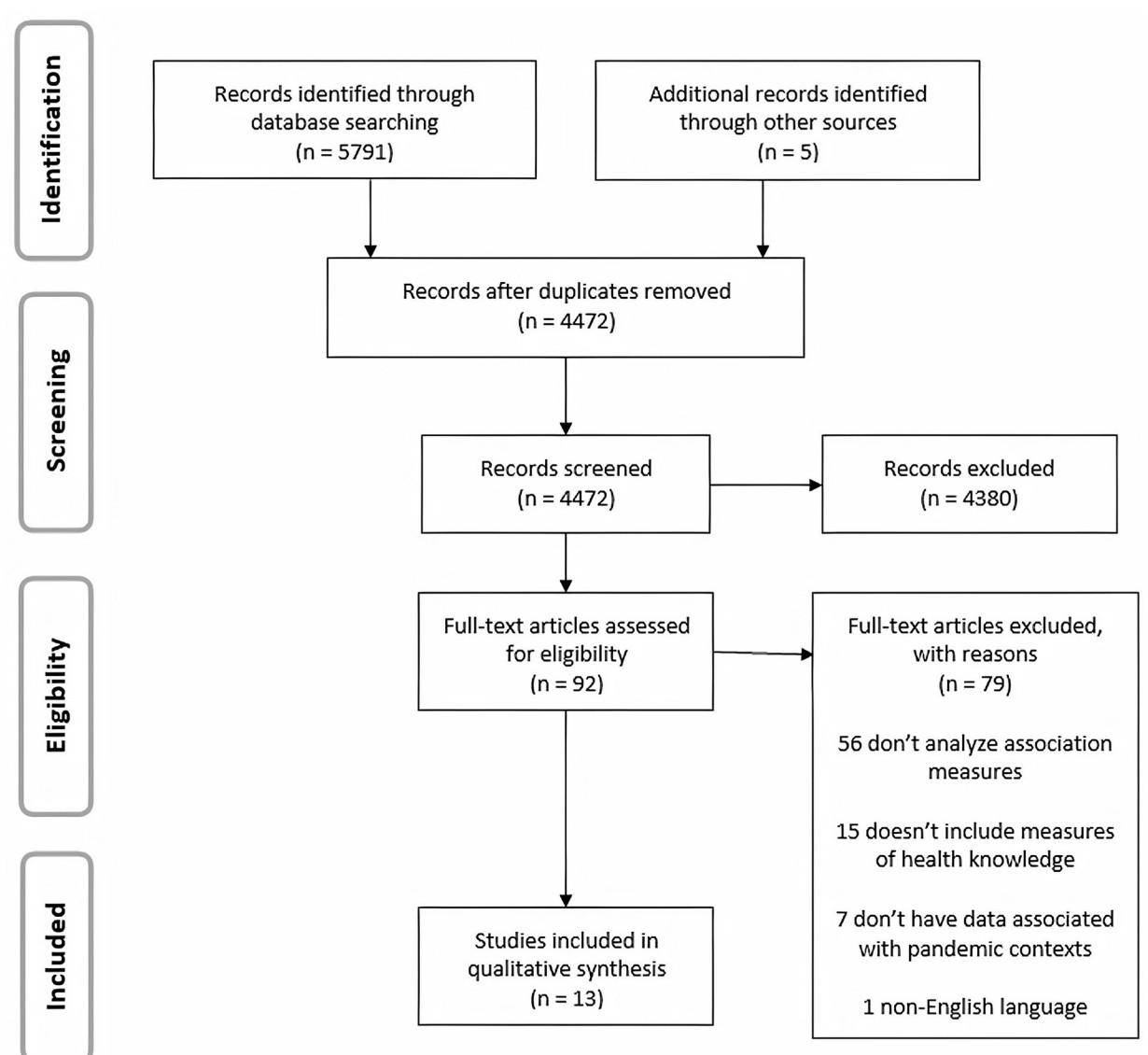

**Fig 1. PRISMA flow diagram of the study selection process.**

inclusion criteria, leaving 13 eligible articles. The complete study selection process is presented in Fig 1.

## Studies' characteristics

The 13 eligible studies examined the relationship between health knowledge and health behaviors or attitudes during pandemics. 13 different population cohorts were analyzed in countries such as Bangladesh, China, Egypt, Hong Kong, India, Iran, Malaysia, United States, Saudi Arabia and Singapore. A total of 26099 peoples were interviewed, grouped into cohorts of health workers (n = 2683), university students (n = 4194), clinical patients (n = 3160), and the general population (n = 16062). All studies adopted a cross-sectional design and were conducted

during three pandemics: Mers-CoV, Influenza A H1N1, and COVID-19. The characteristics and primary results of the studies were extracted and summarized in Table 1.

## Methodological quality assessment

The quality evaluation description for each criterion, as well as the general score is available in S2 Table. Overall, 8 studies had a moderate risk of bias [28–35], while 5 studies showed a high risk of bias [36–40]. Studies with high risk of bias were characterized by unrepresentative samples with respect to target population, whose participants were selected through non-probabilistic convenience sampling [36–40]. Likewise, they did not present response rate and comparability between respondents' characteristics and non-respondents [37–39], and some did not describe measurement instruments' validation process [36, 38, 40]. On the other hand, studies with a moderate risk of bias included representative and satisfactory samples [28–30, 32–34]. Only three of them applied random selection samplings of the sample [29, 30, 34], and described measurement instruments. Only five studies presented validation process [29, 31, 33–35].

## Measure of health knowledge

In all the studies, the measure of health knowledge was quantified through self-report instruments (surveys and questionnaires), and detailed and objective information on the health measures applicable in the pandemics' context was included. All studies evaluated health knowledge as a self-report measure using survey instrument and operationalized it through categories focused on the disease or the virus spread during pandemics, such as: transmission [28–40], prevention [28, 29, 32–40], symptomatology [28, 30, 31, 33–35, 37–39], definition of the pandemic [28, 35, 38, 40], vaccine availability [32–34, 37], infection severity [30, 31, 33, 34], etiology [29, 33, 38], incubation period [33, 38] and communicability period [38]. The frequencies of each category are available in S3 Table.

In general, studies including health workers reported high levels of health knowledge in the face of pandemics, while studies including university students and general population reported variations between high, moderate, and low levels. Thus, two studies found higher knowledge scores about definition, causes and prevention of Covid-19 in a group of doctors (n = 495; M = 38.56; SD = 3.3), nurses (n = 631; M = 37.85; SD = 2.63), paramedics (n = 231; M = 36.72; SD = 4.82) [40], and a sample of clinical dentists (n = 860; M = 8.1; SD = 2.5) [35]. In an academic context, a study found a high level of knowledge about the etiology, definition, transmission, symptoms, prevention, vaccine availability and incubation period of the H1N1 pandemic virus in medical students (n = 125; Resident M = 22.0, SD = 5.8; Fellowship M = 25, SD = 3.4) [33]. And one study reported average level of knowledge about transmission, symptomatology and prevention during the H1N1 pandemic in a group of non-medical students (n = 2882; <Mean score 70.8%; ≥ Mean score 66.3%) [37]. A study examining health knowledge during H1N1 pandemic in a sample of university students (n = 1312) found that medical students reported adequate knowledge about transmission (32.7%), symptoms (53.5%), infection severity (40.4%) and prevention (96.9%) compared with non-medical students (13.7%, 39.6%, 33.7% and 35.0%, respectively) [30].

In the general population, three studies found high levels of health knowledge related to the etiology, transmission and H1N1 prevention in samples of adult participants: (n = 1016; Female, M = 3.60; SD = 0.72 and Male, M = 3.64; SD = 0.76) [29], (n = 1063; 69.7%, SE = 0.5) [28], and about transmission, symptoms and prevention of Covid-19 in a sample of n = 441 adults (M = 6.35, SD = 1.16) [39]. On the other hand, two studies reported mean levels of knowledge about transmission, symptoms, infection severity and H1N1 influenza prevention

**Table 1. Characteristics and primary studies result.**

| Author(s), year and country | Design | Sample | Findings | | Bias Risk |
|---|---|---|---|---|---|
| | | | Primary Outcome | Measure of Association | |
| Almutairi et al., 2015, Saudi Arabia | Cross-sectional | 1147 adults aged >18 years. Male 62%, female 38%. | Health knowledge was a significant predictor for precautionary practices and attitudes in the coronavirus pandemic (MERS-CoV). | Behaviors: $\beta = 0.28$; $SE = 0.02$; $p < 0.001$. | High |
| | | | | Attitudes: $\beta = 0.35$; $SE = 0.08$; $p < 0.001$. | |
| Askarian et al., 2013, Iran | Cross-sectional | 125 resident doctors. The mean age was 30,62 (SD = 5.17) years. Male 59,2%, female 40,8%. | Health knowledge was positively and significantly correlated with health protection practices for pandemic H1N1 influenza. | $r = 0,45$; $p$ value $<0.001$ | Average |
| Etingen et al., 2013, United State | Cross-sectional | 3113 veterans. The mean age was 61,82 (SD = 11.70). Male 96,97%, female 3,03%. | An adequate reception of health information during H1N1 influenza increased the probability of presenting self-protective behaviors in a cohort of older adults | Wear a facemask OR = 1.39, 95% CI 0.99 1.95, $p = 0.053$. | High |
| | | | | Stay home to avoid illness' OR = 0.69, 95% CI 0.53–0.90, $p = 0.006$. | |
| Ho et al., 2013, Singapore | Cross-sectional | 1055 adults aged > 18 years. Male 45,3% and female 54,7%. | Public health knowledge was positively associated with precautionary behavior intentions in the H1N1 pandemic. | $\beta = 0.11$, $p < 0.001$ | Average |
| Keller et al., 2014, China. | Cross-sectional | 2882 university students. Age not reported. Male 70.4%, female 66.6%. | Health knowledge about H1N1 moderately predicted preventive health behaviors. | Wear a facemask: OR = 0.99, 95% IC 0.81–1,20 | High |
| | | | | Increased Hand Washing: OR = 1,10, 95% IC 0.94–1.30 | |
| | | | | Reporting Symptoms OR = 1.10, 95% IC 0.92–1.33. | |
| Krishnappa et al., 2020, India | Cross-sectional | 860 dentists. Continent: Asia 30.7%, Americas 25%, Europe 16.3%, Africa 22.6% and other (Australia and Antarctica 22.6%). | Health knowledge was significantly associated with protective practices in the COVID-19 pandemic. | $r = 0.669$; 95% CI 0,77–26,64 $p <0.05$ | Average |
| Liao et al., 2010, Hong Kong | Cross-sectional | 1016 adults aged >18 years. Male 46%, female 54%. | Health knowledge is a partial mediator between confidence in formal information and personal hygiene practices towards the Influenza A (H1N1) pandemic. | $\beta = 0,35$; ~17% mediation. | Average |
| Lin et al., 2011, China | Cross-sectional | 10669 adults. Aged between 18 and 90 years (M = 41.47 years). Male 45.6%, female 54.4%. | Health knowledge was significantly associated with self-protection practices during the H1N1 pandemic. | OR = 1.57; 95% CI, 1.42 to 1.73; $p<0.0001$ | Average |
| Nabil et al., 2011, Egypt | Cross-sectional | 1312 university students. Male 50.3%, female 49.7%. | Health knowledge in university students was associated with more availability to comply with home quarantine during the H1N1 pandemic. | OR = 0.27, 95% CI 0.2–0.34, $p < 0.001$. | Average |
| Ping et al., 2011, Malaysia | Cross-sectional | 1049 adults aged between 18 and 19 years. Male 37%, female 63%. Malay 41%, Indian 25% and Chinese 34%. | Health knowledge was a significant predictor for the health protective behaviors practice in the three ethnic groups for the pandemic H1N1 outbreak. | Malay: $\beta = 0.08$; $SE = 0.03$; $p < 0.05$; Chinese: $\beta = 0.10$; $SE = 0.03$; $p < 0.001$; Indian: $\beta = 0.08$; $SE = 0.03$; $p < 0.01$. | Average |
| Rahman et al., 2020, Bangladesh | Cross-sectional | 441 adults. 85,7% aged between 18 and 29 years, and 14.3% aged in 30 years or more. Male 68.7% and female 31.3%. | Health knowledge about COVID-19 increases the likelihood of executing preventive practices such as wearing masks and staying home. | Wear a facemask: AOR = 1.54; 95% CI, 1.25 to 1.77; $p<0.01$. | High |
| | | | | Stay home: AOR = 1.73; 95% CI, 1,43 to 2,09; $p<0.01$. | |
| Yap et al., 2010, Singapore | Cross-sectional | 1063 adults aged between 17 and 61 years (M = 21.4; SE = 0.2). Male 95.8%, female 4.2%. Chinese 75.6%, Malay 13.5%, Indian 5.8% and others 2.9%. | High health knowledge was a significant predictor for high levels of protection practices and attitudes towards the Influenza A (H1N1) pandemic. | Practices: $\beta = 0.30$; 95% CI 0.22–0.37; $p < 0.001$. | Average |
| | | | | Attitudes $\beta = 0.21$; 95% CI 0.14–0.28; $p < 0.001$. | |
| Zhang et al., 2020, China | Cross-sectional | 1367 health workers. Male 53.4%, female 46.7%. Doctors 36.5%, Nurses 46.5% and Paramedics 17% | Health knowledge significantly influenced the protective attitudes of health workers. | Attitudes: "Confidence in defeating the virus", OR = 1.41, 95% CI 1.12–1.77, $p < 0.01$.). "Patients must disclose their exposure" OR = 1.217, 95% CI 1.04–1.42, $p < 0.001$. | High |

$\beta$ = beta coefficient, SE = Standard Error, OR: Odds Ratio, AOR = Adjusted Odds Ratio, CI = Confidence Interval.

in samples of adult participants (n = 1055; M = 5.96, SD = 1.61) [34], (n = 1049; M = 7.30, SD = 1.96) [31] during the H1N1 pandemic. A study reported low knowledge about transmission routes of the H1N1 virus in an adult sample (n = 10669) (Coughing and sneezing 75.6%, face to face talk 61.9%, food 30.0%, hand shaking 26.8% and indirect hand contact 22.3%) [35]. And a study found that only a quarter of a sample of n = 3,113 adults reported adequate knowledge about the ways to prevent the H1N1 virus (Stayed home, 22.47%, Wore mask, 18.22% and took antiviral medication, 17.14% [36]. Finally, a study found that a high percentage of an adult sample (n = 1147) know about the MERS-CoV (91.6%). A significant percentage of participants reported inaccurate information about transmission routes (43.9%), infection severity (48.1%), and they stated did not feel confident about the incubation period (50.5%) or communicability period (36.5%) [38].

## Health knowledge and health behaviors

12 studies examined the effect of health knowledge in the adoption of health behaviors in the general population, university students, and health workers during MERS-CoV, H1N1 and Covid-19 pandemics. In the general population, two studies reported that health knowledge in adult participants was positively associated with self-protection and precautionary practices (n = 10669; OR = 1.57; 95% CI, 1.42 to 1.73, $p < 0.0001$ and n = 1055; $\beta = 0.11$, $p < 0.001$, respectively) [32, 34]. Likewise, two studies found that health knowledge behaved as significant predictor factor for performance of protective behaviors (n = 1063; $\beta = 0.30$; 95% CI 0.22–0.37; $p < 0.001$) [28] and precaution (n = 1147; $\beta = 0.28$; SE = 0.02; $p < 0.001$) [38], such as: washing hands, wearing a mask, covering when sneezing or coughing, monitoring temperature, avoiding public places, and seeking medical help. One study found that health knowledge predicted protective behaviors (mask use, hygiene practices, physical distancing, avoiding crowds, taking preventive medicine, and adopting healthy lifestyles) in three different ethnic groups: Malay (n = 435; $\beta = 0.08$; SE = 0.03; $p < 0.05$); Chinese (n = 352; $\beta = 0.10$; SE = 0.03; $p < 0.001$) and Indian (n = $\beta = 0.08$; SE = 0.03; $p < 0.01$) [31].

In addition, two studies found health knowledge increases probability of preventive practices both in clinical settings (n = 3113; wear a face mask OR = 1.39, 95% CI 0.99 1.95, p = 0.053 and stay home to avoid illness' OR = 0.69, 95% CI 0.53–0.90, p = 0.006) [36] or in non-clinical settings (n = 441) use of masks: AOR = AOR = 1.54; 95% CI, 1.25 to 1.77; p <0.01 and staying home: AOR = 1.73; 95% CI, 1.43 to 2.09; p <0.01) [39]. Related to the role health knowledge played in sensitization processes by government in the general population during H1N1 pandemic emergency, a study reported that health knowledge acted as a significant mediator (n = 1016; $\beta = 0.35$; ~ 17% mediation) between reliance on formal health information and the adoption of hygiene practices such as: washing hands, covering when sneezing or coughing, and discarding items used to sanitize objects [29].

One study in academic context found that high levels of health knowledge were positively associated with health behaviors, such as hygiene habits, use of Biosafety elements, self-protection behaviors, and social distancing in a sample of n = 125 medical resident students (r = 0.45; p-value <0.001) [33]. Furthermore, one study reported health knowledge has favored compliance with home quarantine in a sample of n = 1312 medical and non-medical students (OR = 0.27, 95% CI 0.2–0.34, p <0.001) [30]. One study found that in a sample of university students (n = 2882), health knowledge played an important role in preventive health behaviors, such as masks use (OR = 0.99, 95% CI 0.81–1.20), hand washing (OR = 1.10, 95% CI 0.94–1.30), and symptom report (OR = 1.10, 95% CI 0.92–1.33) [37].

On the other hand, in the clinical context, two studies found that high levels of health knowledge in samples of n = 331 health workers and n = 860 dentists were positively associated

with health behaviors such as masks use, hygiene habits, avoid crowds, application of influenza vaccines, sensitize patients about prevention measures, record symptoms and discuss infection risk (r = 0.28; p-value <0.01 and r = 0.66; p <0.05, respectively) [28, 35]. In these results, the findings show that health knowledge influenced health behaviors in different population groups. However, the level of knowledge or work experience of health workers and university medical students notably favored the execution of prevention and self-protection practices for both personal and collective care (especially towards their patients) [28, 33, 35] compared to the general population [28, 29, 31, 32, 34, 36, 38, 39].

## Health knowledge and health attitudes

Three studies explored the influence of health knowledge in the acquisition of health attitudes in two population groups during MERS-CoV, H1N1 and Covid-19 pandemics [28, 38, 40]. In general population, Yap et al. (2010) [28] associated the levels of knowledge (definition, transmission, prevention, and symptoms) and attitudes related to preventive measures in a cohort of adults during the H1N1 pandemic. It was observed that high levels of health knowledge in the participants allowed them to have better attitudes ($\beta$ = 0.21; 95% CI 0.14–0.28; $p < 0.001$), reflecting adequate perceptions about the implementation of hygiene and self-protection practices, the importance of influenza vaccination, the role of medical services and the risk of infection. Likewise, Almutairi et al., (2015) [38] examined the influence of knowledge on health and attitudes related to the coronavirus pandemic (MERS-CoV). The authors found significant predictive values that linked the health knowledge of a group of Saudi adults with more adequate attitudes towards the pandemic ($\beta$ = 0.35; SE = 0.08; $p < 0.001$). These were related to correct perceptions about the severity of the disease, the health measures proposed by the government and the availability to comply with social restrictions.

For their parte, Zhang et al., (2020) [40] explored the link between health knowledge and health attitudes in a group of health workers facing the COVID-19 pandemic. The findings allowed observing that high levels of health knowledge in health workers favored their attitudes associated with the level of confidence in defeating the virus and the forms of prevention in patients affected during the COVID-19 pandemic emergency (OR = 1.41, 95% CI 1.12–1.77, $p < 0.01$; OR = 1.217, 95% CI 1.04–1.42, $p < 0.001$). In three studies, the authors found health knowledge acted as a predictor of attitudes associated with health issues during pandemics in two population groups. A study including health workers [40] found some factors such as work experience, risk exposure and level of knowledge had a notable influence on their health attitudes compared to the two studies that included general population [28, 38]

## Discussion

This systematic review identified 13 eligible studies that evaluated measures of knowledge, behavior, and health attitudes in adult cohorts during pandemic outbreaks of MERS-CoV, H1N1, and COVID-19. The synthesis result of the evidence suggests that health knowledge has a significant effect on the acquisition of health behaviors and attitudes, reflecting the efforts of the health system, the government and the community to face the negative impact of pandemic emergencies. Thus, these results show people with adequate knowledge of health perform good preventive practices and present appropriate perception of health emergency, which favor the consolidation of effective rapid responses to face the risk of pandemic infection.

In all the studies, health knowledge was approached as a self-report measure, where the information that the participants had regarding health issues in pandemic situations was evaluated. The wide conceptual heterogeneity of the health knowledge variable was evident in the

synthesis of the results, however, most of the studies included central categories: transmission, symptoms, etiology, infection severity and prevention [33, 35, 38]. Although in public health research, the application of undifferentiated and arbitrary meanings to the term health knowledge has generated conceptual confusions that represent a significant source of bias [1], from the findings of this review it is evident that health knowledge has been approached from categories that report the characteristics of the disease (definition, etiology, transmission, prevention, symptoms), excluding positive health factors, such as: lifestyles, healthy habits, protective factors, rights and provision of health services. A less pathological approach and with a greater emphasis on positive health aspects could broaden the notions of the health of the public and its effect of interaction with changes in people's behavior could bring beneficial effects for the public health of society.

The effect of health knowledge on the adoption of health behavior was explored in most of the studies. The results indicate that health knowledge is a predictor and mediator of hygiene practices, self-protection behaviors, restriction and social distancing or seeking medical help in all cohorts during pandemic contexts. Four were the health behaviors with the highest prevalence in the study's general synthesis: hand washing, mask's use, covering up the face when sneezing or coughing, and social isolation, finding an increase in the execution of these practices when the participants reported good knowledge in health [28, 32]. This interaction effect between health knowledge and changes in people's behavior can be explained by the practical nature, easy access and simplicity in the execution of these practices, which could facilitate their adherence to lifestyles and participants' individual hygiene habits. During the early stages of pandemic outbreaks (H1N1 and Covid-19), people who reported high levels of health knowledge on aspects such as transmission, symptomatology, and prevention adopted these measures as rapid responses for control and surveillance of outbreaks massive infection [29, 31, 39]. Previous evidence has shown that the timely implementation of these practices significantly reduces the spread of highly pathogenic viruses [41, 42].

Only three studies found a significant effect of health knowledge on participants' health attitudes. All three studies provide some evidence that health knowledge may be a predictor of health attitudes during pandemic outbreaks [28, 38, 40]. Furthermore, the evidence suggests that adequate health knowledge helps correct misconceptions and myths about the disease, as well as increasing the perception of susceptibility against the infection risk [28, 38]. These attitudes were characterized as being intentional, evaluative, subjective, and cognitive in nature, reflecting perceptions and thoughts about government measures to contain outbreaks, the risk of infection, and evaluative judgments about exposure to the disease. These findings may explain the effect of acquiring correct health information and the selection of legitimate information sources for an adequate understanding of the circumstances of a pandemic emergency, especially those associated with health care and prevention [43].

## Limitations

To consider the evidence provided in this rapid review, is important to take account of its limitations. Initially, the cross-sectional nature of all the studies may limit the validity and generalization of the findings. Second, a significant heterogeneity of the data was found, due to the methodological and conceptual variation used in the studies. The reasons that could explain these limitations is the challenge of obtaining reliable estimates, with a high degree of evidence, in the context of a pandemic. In this way, the non-random sampling of the sample, the use of self-report scales and questionnaires, and the use of virtual means for the measurement of the variables can represent an important source of bias. However, this rapid review of the evidence found a substantial number of studies that empirically substantiate the effects of health

knowledge on health behaviors and attitudes during public health emergencies caused by pandemics.

## Conclusions

The findings of this rapid review indicate an association between health knowledge and the adoption of health behaviors and attitudes. However, the data nature to examine the effects of their interaction remains challenging in pandemic settings. The concept of health knowledge needs to be extrapolated from mere pandemic knowledge and must incorporate clear and detailed information on general and specific aspects of health, linked to adequate values, intentions and perceptions about personal and collective care in the midst of pandemics. Therefore, a clear, consistent and reliable knowledge on health issues would help people to understand and adopt rapid behavioral and cognitive responses to face a pandemic. Also, would allow the development of healthy public policies, based on communities' experience that highlight the capacity for people commitment and the actions' efficiency of health promoters, researchers and government entities to mitigate risks and harmful factors for health. On the other hand, an adequate conceptual construction could be of utmost importance to address the contribution of health knowledge in emergency situations. Thus, for future research, one could try to adopt general and specific categories in health, including aspects associated with health care and disease prevention, to guarantee the homogeneity and generalization of empirical results. Likewise, could favor the adequate transmission of health information, also representing an effective strategy to control the massive spread of highly pathogenic viruses.

## Supporting information

**S1 Table. Inclusion and exclusion criteria for article selection.**
(DOCX)

**S2 Table. Quality assessment of included studies.**
(DOCX)

**S3 Table. Operationalization of the health knowledge measurement.**
(DOCX)

**S1 Appendix. PRISMA checklist.**
(DOC)

**S2 Appendix. Search criteria.**
(DOCX)

## Author Contributions

**Conceptualization:** Fabio Alexis Rincón Uribe.

**Data curation:** Fabio Alexis Rincón Uribe, Rejane Célia de Souza Godinho, Marcos Antonio Santos Machado, Janari da Silva Pedroso.

**Formal analysis:** Fabio Alexis Rincón Uribe, Rejane Célia de Souza Godinho, Marcos Antonio Santos Machado, Cristian Ariel Neira Espejo, Natalia Carvalho Vianna de Sousa, Leonardo Lucas de Sousa, Marcos Vinicius Monteiro Barbalho, Janari da Silva Pedroso.

**Funding acquisition:** Kleber Roberto da Silva Gonçalves Oliveira, Pedro Paulo Freire Piani, Janari da Silva Pedroso.

**Investigation:** Fabio Alexis Rincón Uribe, Rejane Célia de Souza Godinho, Marcos Antonio Santos Machado, Cristian Ariel Neira Espejo, Natalia Carvalho Vianna de Sousa, Leonardo Lucas de Sousa, Marcos Vinicius Monteiro Barbalho, Pedro Paulo Freire Piani, Janari da Silva Pedroso.

**Methodology:** Fabio Alexis Rincón Uribe, Rejane Célia de Souza Godinho, Marcos Antonio Santos Machado, Cristian Ariel Neira Espejo, Natalia Carvalho Vianna de Sousa, Leonardo Lucas de Sousa, Marcos Vinicius Monteiro Barbalho, Pedro Paulo Freire Piani, Janari da Silva Pedroso.

**Project administration:** Fabio Alexis Rincón Uribe.

**Resources:** Fabio Alexis Rincón Uribe, Kleber Roberto da Silva Gonçalves Oliveira, Cristian Ariel Neira Espejo, Pedro Paulo Freire Piani.

**Supervision:** Fabio Alexis Rincón Uribe, Kleber Roberto da Silva Gonçalves Oliveira.

**Visualization:** Fabio Alexis Rincón Uribe.

**Writing – original draft:** Fabio Alexis Rincón Uribe.

**Writing – review & editing:** Fabio Alexis Rincón Uribe.

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
