## [Decision Letter · Decision Letter 0]

7 Jan 2021

PONE-D-20-31471

Effects of health knowledge in the health behaviors and attitudes adoption during pandemic emergencies: a rapid review

PLOS ONE

Dear Dr. Fabio Alexis Rincon Uribe,

Thank you for submitting your manuscript to PLOS ONE. After careful consideration, we feel that it has merit but does not fully meet PLOS ONE’s publication criteria as it currently stands. Therefore, we invite you to submit a revised version of the manuscript that addresses the points raised during the review process.

We look forward to receiving your revised manuscript.

Kind regards,

Billingsley Kaambwa

Academic Editor

PLOS ONE

Journal Requirements:

2. Please consider modifying your title to ensure that it is specific, descriptive, and concise (for example by specifying this is a systematic review).

Reviewers' comments:

Reviewer's Responses to Questions

**Comments to the Author**

1. Is the manuscript technically sound, and do the data support the conclusions?

Reviewer #1: Yes

Reviewer #2: Yes

2. Has the statistical analysis been performed appropriately and rigorously? 

Reviewer #1: N/A

Reviewer #2: N/A

3. Have the authors made all data underlying the findings in their manuscript fully available?

Reviewer #1: Yes

Reviewer #2: Yes

4. Is the manuscript presented in an intelligible fashion and written in standard English?

Reviewer #1: Yes

Reviewer #2: Yes

5. Review Comments to the Author

Reviewer #1: Valuable insights provided by your paper, and I commend you on doing this work. I feel the paper could do with a minor round of revision, especially for the abstract section. For example, in line 27, the author wrote “… to carry out a literature quick review …”. Consider reordering the words. I feel there is not enough information in the result section to help the reader understands how the authors have reached to make the claims in lines 40-41. In line 43, the authors wrote “… these measures …”. I find this confusing as it is not not clear 'which measures' the authors were referring to. I also feel that in lines 44-45, the authors did not provide any information in the abstract section do demonstrate this claim. In line 57, there is typo – "offered1".

Reviewer #2: The introduction is well written but I highly suggest to integrate the introduction and statement of the problem. There are some issues that needs to be addressed in the methodology section such as inclusion of studies that report only significant results, how sound the ONS cut off points are, how interaction was done in the absence of meta-analysis. In the results section, there are some points that needs clarification on the description of the studies, methodological quality assessment. There is considerable typological issues. Hence, the manuscript needs a careful proof reading to address these issues.

6. PLOS authors have the option to publish the peer review history of their article (what does this mean?). If published, this will include your full peer review and any attached files.

Reviewer #1: No

Reviewer #2: **Yes: **Dr Fisaha Haile Tesfay

---

## [Author Response · Author response to Decision Letter 0]

19 Feb 2021

February 19, 2021

Dear Plos One Publishing Group

We appreciate the suggestions and comments made on our manuscript. Below, we answer the editor and reviewers’ comments. Modifications made in the manuscript are in red. 

Response to Editor

R/: We ensure that our manuscript meets PLOS ONE style requirements for editing and presentation. 

2. Please consider modifying your title to ensure that it is specific, descriptive, and concise (for example by specifying this is a systematic review).

R/: We have accepted modifying the title to make it more specific and concise (In line 1 – 2).

Response to Reviewers

Commentaries

Reviewer 1

1. The author wrote “… to carry out a literature quick review …”. Consider reordering the words (In line 27). 

R/: We have rearranged this sentence: The main objective of this study was to conduct a rapid systematic review.

2. I feel there is not enough information in the result section to help the reader understands how the authors have reached to make the claims. .

R/: We have included statistical data in the results section to support claims made in the conclusions.

3. In line 43, the authors wrote “… these measures …”. I find this confusing as it is not clear 'which measures' the authors were referring to.

R/: In this line, we refer to preventive measures presented in the evidence; therefore, we have rectified this sentence (In line 43). 

4. I also feel that in lines 44-45, the authors did not provide any information in the abstract section do demonstrate this claim. 

R/: We have modified this statement, to link it with information in the results section of the summary (In line 44 – 45). 

5. In line 57, there is typo – "offered1". 

R/: This typo mistake was corrected. (line 57)

Reviewer 2

1. PRISMA P is for protocol. Should be PRISMA. 

R/: PRISMA-P checklist was changed to PRISMA (In line 31). 

2. Can you explain what a solid knowledge meant? 

R/: We want to indicate that health knowledge based on clear and objective information could facilitate rapid adoption of responses to a pandemic. In this way, we have decided to improve this statement and detail influence of a good health knowledge on people's rapid responses to a pandemic (In line 44 - 46). 

3. Can you add a bit on how health knowledge is an objective construct? 

R/: We added a brief explanation about how health knowledge is a construct of an objective nature. (In line 50 – 52)

4. Public health research or health knowledge? 

R/: In this section, the idea focuses on the inconsistencies found in health knowledge research during pandemics. We corrected this sentence to specifically indicate this idea. (In line 68-69)

5. I will highly suggest to merge and integrate this with the rest of the introduction.

R/: The suggestion of reviewer 2 was accepted and we included this section in manuscript general introduction. (In line 88 – 95, 100-101 and 106 - 109)

6. Change this to PRISMA . 

R/: PRISMA-P checklist was replaced by PRISMA. (In line 113) Also, bibliographic reference was replaced (lines 476-478).

7. Methodological guide like what? 

R/: This section indicates the methodological guide called "The Interim Guidance from the Cochrane Rapid Reviews Methods Group" was used. We restructured this sentence to clarify the statement (In line 113 – 115). 

8. What about the none significant reports? Your inclusion criteria will definitely lead to publication bias. Addressing this issue is important 

R/: We reformulated this criterion, emphasizing the main interest of the systematic review, which is to examine the influence of health knowledge on the adoption of health behaviors and attitudes of people in the midst of pandemics (In line 126 -127) 

9. Good to put a good reason on why your review was limited to English language only. 

R/: Our main reason to not include studies in other languages was implementation of the methodology for the preparation of rapid systematic reviews, proposed by Cochrane Group. In this methodology, review processes are accelerated, and some stages are limited, to offer relevant and substantial conclusions in emergency situations.

10. Is study selection different from inclusion and exclusion criteria? 

R/: We reviewed methodology about systematic reviews and integrated these two sections as one phase of the process. During study selection, inclusion and exclusion criteria are activated by authors to define the number of relevant articles that will be included in the review (In line 126 – 140). 

11. Instead of saying statistical significance, it is good if you rephrase it as statistical estimates 

R/: Considering the suggestion of Reviewer 2, the expression statistical significance was reformulated as statistical estimates (In line 146). 

12. Normally, the ONS doesn't provide such as cut off point. What we usually do when we use the ONS is make relative comparison. So, check that please. 

R/: We reviewed this section and made a relative comparison with the cutoff points established by previous systematic reviews studies. In this way, we based the cutoff points according to this literature (In line 154 – 155). 

13. If you didn't do meta-analysis, it is not clear how interaction was assessed here? 

R/: This section was reformulated to explain in detail the synthesis and presentation process’ results (In line 161 – 167). 

14. Clarify what manual searching refers to. 

R/: Two types of manual searches were included in this review (In line 169 – 171)

15. After removing duplicates. 

R/: This sentence was rewritten, including organization suggested by reviewer 2 (In line 171).

16. Screening. 

R/: Expression "full-text selection" was replaced by the word “screening” (In line 172). 

17. Remove proposed. 

R/: Word “proposed” was removed (In line 157, in the first version of the manuscript).

18. This seems duplication because it is already mentioned above. 

R/: To avoid duplicate information in the text, sentence “with a publication date that varies between 2009 and 2020” was removed (In line 157, in the first version of the manuscript), since this information has already been mentioned in the method (in lines 135).

19. This is not clear. Rephrase it please. 

R/: The phrase was rewritten (In line 182-183). 

20. Good to provide a detailed description of what is missing in the studies with high risk of bias.

R/: In this section, we included a detailed description about was missing in the studies at high risk of bias (In line 187-195). 

21. This is not clear which for the studies with the highest risk and which is for the lowest risk.

R/: We made a clearer division between studies with high and moderate risk of bias (In line 187– 188).

22. No comparison was made between the various groups such as students, health workers and other groups when it comes to the level of knowledge. 

R/: We made a section in the results to report levels of knowledge between three groups examined in the studies. However, due to data heterogeneity, it was not possible to do a strict comparison (In line 209 – 241). 

23. How many studies measured health knowledge as a continuous measure and how many measured operationalize through categories. 

R/: We reviewed this sentence and found an error in the use of concept “continuous measure”. In this way, we reformulated this sentence and indicated the measure used to assess health knowledge. Thus, all studies included in the review evaluated health knowledge as a self-report measure, through categories associated with the pandemic outbreak (In line 202 – 203).

24. Same here in terms of comparing the various groups 

R/: We accomplished a narrative comparison between groups, based on the results obtained from studies (In line 243 – 285). 

25. Add the magnitude of the estimates? 

R/: The magnitudes of statistical estimates were added in the results.

26. What was the magnitude of the estimate? 

R/: The magnitudes of the statistical estimates were included in the results.

27. Same here (what is moderate prediction?) 

R/: The phrase that explains statistical magnitudes was reorganized, to not use a statistical discourse that could be ambiguous (in line 272 - 275).

28. Good to show the magnitude of the association. 

R/: The magnitudes of the association were included. In addition, the other magnitudes were included to support the results with their statistical data (In line 292, 298 and 305 – 306).

29. Is it only among health workers? 

R/: Although we cannot make a comparison in statistical terms, which would determine whether there is a difference in health attitudes among health workers and the general population, we have considered making a narrative comparison between two groups based on the extracted results in studies (In line 306 – 310). 

30. It is not clear what predictive and mediating effect mean here? 

R/: We clarified how good health knowledge influences acquisition of health behaviors and attitudes. For that, we modified the discourse to not exceed statistical terminology, which was already exposed in the results section (In line 321 – 319). 

31. It is not clear how health knowledge is continuous measure and in the results while it was categorized as transmission, symptoms, etiology and infection severity. 

R/: We found a conceptual error when using this term (continuous measure), for this reason we reformulated this sentence. Thus, health knowledge was operationalized as a self-report measure, and people were directly asked about their knowledge about different categories associated with the pandemic and prevention measures (In line 320). 

32. It is very important to define health knowledge, health behavior, and attitude clearly in the methods section. 

R/: In the methods section, we defined health knowledge, behaviors, and health attitudes (In line 127 – 135). 

Additional changes

1. We updated institutional affiliations of some authors and highlighted changes in red (In line 14 – 17). 

2. We have updated our financial disclosure and included it in the cover letter.

---

## [Decision Letter · Decision Letter 1]

16 Aug 2021

Health knowledge, health behaviors and attitudes during pandemic emergencies: a systematic review

PONE-D-20-31471R1

Dear Dr. Rincon Uribe,

We’re pleased to inform you that your manuscript has been judged scientifically suitable for publication and will be formally accepted for publication once it meets all outstanding technical requirements.

Kind regards,

Cho Naing, MBBS, PhD, FRCP

Academic Editor

PLOS ONE

Additional Editor Comments (optional):

The authors have addressed the comments provided by the reviewers.

Still, the authors need to update the usage of terms.

E.g PRISMA checklist, rather than PRISMA group parameters.

These minor things can be corrected during a prof reading.

Thank you

Reviewers' comments:

Reviewer's Responses to Questions

**Comments to the Author**

1. If the authors have adequately addressed your comments raised in a previous round of review and you feel that this manuscript is now acceptable for publication, you may indicate that here to bypass the “Comments to the Author” section, enter your conflict of interest statement in the “Confidential to Editor” section, and submit your "Accept" recommendation.

Reviewer #1: All comments have been addressed

2. Is the manuscript technically sound, and do the data support the conclusions?

Reviewer #1: Yes

3. Has the statistical analysis been performed appropriately and rigorously? 

Reviewer #1: N/A

4. Have the authors made all data underlying the findings in their manuscript fully available?

Reviewer #1: Yes

5. Is the manuscript presented in an intelligible fashion and written in standard English?

Reviewer #1: Yes

6. Review Comments to the Author

Reviewer #1: Dear Author,

Thank you for your revision. You have addressed the relevant issues raised in the first round of the paper. However, I feel you should give a good grammatical editing to your paper if you are presented with the opportunity to make final changes. There are few grammatical errors in the paper. Nevertheless, the paper makes an important contribution to the importance of health knowledge in a pandemic context.

7. PLOS authors have the option to publish the peer review history of their article (what does this mean?). If published, this will include your full peer review and any attached files.

Reviewer #1: No

---

## [Editor Report · Acceptance letter]

26 Aug 2021

PONE-D-20-31471R1 

Health knowledge, health behaviors and attitudes during pandemic emergencies: a systematic review 

Dear Dr. Rincón Uribe:

I'm pleased to inform you that your manuscript has been deemed suitable for publication in PLOS ONE. Congratulations! Your manuscript is now with our production department. 

Kind regards, 

on behalf of

Professor Cho Naing 

Academic Editor

PLOS ONE